# Senescent Macrophages Release Inflammatory Cytokines and RNA-Loaded Extracellular Vesicles to Circumvent Fibroblast Senescence

**DOI:** 10.3390/biomedicines12051089

**Published:** 2024-05-14

**Authors:** Camille Laliberté, Bianca Bossé, Véronique Bourdeau, Luis I. Prieto, Genève Perron-Deshaies, Nhung Vuong-Robillard, Sebastian Igelmann, Lisbeth Carolina Aguilar, Marlene Oeffinger, Darren J. Baker, Luc DesGroseillers, Gerardo Ferbeyre

**Affiliations:** 1Département de Biochimie et Médecine Moléculaire, Université de Montréal, Montréal, QC H3C 3J7, Canadabianca.bosse@umontreal.ca (B.B.); veronique.bourdeau@umontreal.ca (V.B.); 2Centre de Recherche du Centre, Hospitalier de l’Université de Montréal (CRCHUM), Montréal, QC H2X 0A9, Canadageneve.perron-deshaies.chum@ssss.gouv.qc.ca (G.P.-D.);; 3Department of Biochemistry and Molecular Biology, Mayo Clinic, Rochester, MN 55905, USA; prieto.luis@mayo.edu (L.I.P.); baker.darren@mayo.edu (D.J.B.); 4Department of Pediatrics, Mayo Clinic, Rochester, MN 55905, USA; 5Laboratory of Cellular Metabolism and Metabolic Regulation, Department of Oncology, KU Leuven and Leuven Cancer Institute (LKI), 3000 Leuven, Belgium; 6Institut de Recherches cliniques de Montréal (IRCM), 110 Avenue des Pins Ouest, Montréal, QC H2W 1R7, Canadamarlene.oeffinger@ircm.qc.ca (M.O.); 7Paul F. Glenn Center for Biology of Aging Research, Mayo Clinic, 200 1st ST SW, Rochester, MN 55905, USA

**Keywords:** senescence, macrophages, extracellular vesicles, cancer

## Abstract

Senescent cells, which accumulate with age, exhibit a pro-inflammatory senescence-associated secretory phenotype (SASP) that includes the secretion of cytokines, lipids, and extracellular vesicles (EVs). Here, we established an in vitro model of senescence induced by Raf-1 oncogene in RAW 264.7 murine macrophages (MΦ) and compared them to senescent MΦ found in mouse lung tumors or primary macrophages treated with hydrogen peroxide. The transcriptomic analysis of senescent MΦ revealed an important inflammatory signature regulated by NFkB. We observed an increased secretion of EVs in senescent MΦ, and these EVs presented an enrichment for ribosomal proteins, major vault protein, pro-inflammatory miRNAs, including miR-21a, miR-155, and miR-132, and several mRNAs. The secretion of senescent MΦ allowed senescent murine embryonic fibroblasts to restart cell proliferation. This antisenescence function of the macrophage secretome may explain their pro-tumorigenic activity and suggest that senolytic treatment to eliminate senescent MΦ could potentially prevent these deleterious effects.

## 1. Introduction

Cellular senescence is an important cellular mechanism underlying aging and age-related diseases [1]. Senescence can be defined as a stable cell cycle arrest in response to endogenous or exogenous stresses, such as telomere dysfunction, DNA damage, oncogene activation, oxidative stress, or mitochondrial dysfunction [2]. Typically, these activate the p53/p21 and p16/RB pathways, which actively inhibit the growth of damaged cells, ultimately thwarting the development of cancer [3]. Senescent cells secrete pro-inflammatory molecules such as cytokines, chemokines, growth factors, and proteases. This trait is called the senescence-associated secretory phenotype (SASP) [4]. The SASP is thought to alert the immune system, which allows the elimination of damaged cells [5]. However, if senescent cells are not eliminated, SASP factors such as growth factors and proteases contribute to angiogenesis and tumor invasion [6,7]. Senescent cells accumulate in aging organisms [8,9,10,11] and are linked to age-related diseases, such as pulmonary fibrosis [12,13], type 2 diabetes, [14] and osteoarthritis [15,16,17]. When senescent cells accumulate, the persistence of their secretions favors chronic inflammation, which is characterized by infiltration of immune cells in tissues leading to fibrosis and necrosis [18]. Chronic inflammation is an important characteristic of aging [19] and has been also linked with age-related diseases [20,21]. More importantly, the elimination of senescent cells extends the healthy lifespan of old mice and prevents some age-related conditions [22,23].

Several studies show that senescence in macrophages could play a role in the development of aging-associated diseases. For instance, angiogenic secretions of senescent MΦ could contribute to age-related macular degeneration [24]. Additionally, by secreting pro-inflammatory factors and metalloproteases, senescent MΦ that accumulate in the atheromatous plaque favor the progression of atherosclerosis [25]. Senescence in microglia—specialized MΦ of the central nervous system—has been reported to participate in tau protein aggregation, which contributes to neurodegenerative diseases [26]. In addition, senescent MΦ support neoplastic progression via their effects in the tissue microenvironment [27,28]. Intriguingly, it has been suggested that macrophages account for a large proportion of cells bearing the senescent markers p16 and senescence-associated β-Galactosidase in mice (SA-β-Gal) [29]. However, the biology of senescent MΦ is still poorly understood.

In addition to soluble factors, senescent cells secrete large amounts of extracellular vesicles (EVs) [30,31,32,33,34,35,36,37,38]. EVs are considered important players in intercellular communication, transporting a wide range of biological molecules, including proteins, lipids, nucleic acids, metabolites, and organelles, such as mitochondria [39,40,41,42,43,44,45,46,47]. EVs are mainly divided into two major categories: microvesicles and exosomes. Microvesicles are typically larger vesicles (100 nm to 1 µm), and they bud directly from the plasma membrane [37,48]. Exosomes are smaller vesicles (30–150 nm) that are formed in endosomes. In a similar manner to soluble secretions, EVs affect the producing cell in an autocrine fashion or their local environment through paracrine signaling [49,50]. Interestingly, the presence of EVs in blood circulation suggests they can participate in endocrine signaling [51]. EVs from senescent cells seem to have pathological effects, including promoting cancer, decreasing bone regeneration capacity, and inducing paracrine senescence [32,35,52]. It has been proposed that senescent-derived EVs are part of the SASP [53].

Here, we cataloged the transcriptome changes associated with senescence in macrophages, as well as the RNA and protein content of their EV secretions. Moreover, we found that together, the secretory products of macrophages promote escape from senescence, a process associated with early stages of tumorigenesis.

## 2. Materials and Methods

### 2.1. Animals

The p16-3MR mice were generously provided by Dr. Judith Campisi and maintained by the animal facility of the CRCHUM using protocols approved by the Institutional Animal Care Committee (IACC) of the CRCHUM, which follows the conditions and guidelines established by the Canadian Council on Animal Care (CCAC).

### 2.2. Cell Culture

Cells were cultured in Dulbecco’s modified Eagle medium (DMEM, Wisent, Saint-Jean-Baptiste, QC, Canada), supplemented with 10% fetal bovine serum (Gibco), 2 mM L-glutamine (Wisent), and 1% penicillin G/streptomycin sulfate (Wisent). RAW 264.7 cells were obtained from ATCC and detached mechanically, whereas Phoenix cells (a kind gift from Dr. S.W. Lowe; Memorial Sloan-Kettering Cancer Center, New York, NY, USA) were detached using trypsin (Wisent).

### 2.3. Bone Marrow-Derived Macrophages Extraction and Senescence Induction

The p16-3MR mice were euthanized according to the ethics committee protocol of the CRCHUM. Mice femurs were then flushed with DMEM, and the bone marrow was mechanically homogenized using a 21G needle. Red blood cells were lysed using 1X RBC lysis buffer (eBiosciences, San Diego, CA, USA) and the remaining cells were washed and put in a 10 cm petri dish with regular media overnight to remove the adherent cells. The next day (day 0), non-adherent cells were counted and seeded at a density of 1 × 10^6^ cells/mL in differentiation media [DMEM supplemented with 10% fetal bovine serum (Gibco), 2 mM L-glutamine (Wisent), 1% penicillin G/streptomycin sulfate (Wisent), and 25 ng/μL recombinant mouse macrophage colony stimulating factor (M-CSF, BioLegend, San Diego, CA, USA)]. Senescence was induced by treating the cells with 500 µM H_2_O_2_ for 2 h on day 3 of differentiation. After this treatment, the media was changed with fresh differentiation media. Senescence was evaluated at the end of differentiation on day 6.

### 2.4. Viral-Mediated Gene Transfer

Plasmids used were MSCV hygro ∆Raf-1:ER and MSCV hygro ER (which were obtained by subcloning BamHI/SnaBI from pBABE-∆Raf-1:ER or pBABE-ER, gifts from S.W. Lowe, into MSCV hygro with modified multiple cloning sites), as well as MSCV ∆Raf-1:ER IRES tdTomato (obtained by subcloning the insert of MSCV hygro ∆Raf-1:ER with BamHI/XhoI into BglII/XhoI of MSCV-IRES_Tomato from Addgene, plasmid #107229). A day before transfections, Phoenix Ampho packaging cells were seeded to obtain approximately 50% confluency in 10 cm plates. Transfections were performed using calcium phosphate precipitation with 20 µg plasmid DNA and 10 µg of amphotropic accessory plasmid. To increase viral production, cells were treated with 10 mM sodium butyrate the day after transfection. Medium was replaced 8 h later. The next day, the culture medium containing viral particles was harvested and filtered (0.45 µm), then supplemented with 10% fresh medium and 4 µg/mL polybrene. This medium with viral particles was transferred on RAW 264.7 for infection. Fresh medium was added to transfected Phoenix cells, which were incubated at 37 °C for 6 h before repeating the infection protocol. Selection of infected RAW 264.7 cells was carried out according to the selectable gene of the plasmid used for viral-mediated gene transfer using either hygromycin B (70 µg/mL, 10 days) or via sorting by FACS at IRIC cytometry platform on a BD FACSAria cell sorter for cells exhibiting a high tdTomato signal.

### 2.5. Growth Curve

The relative growth of RAW 264.7 cells was assessed by crystal violet retention assay. RAW 264.7 cells expressing ∆Raf-1:ER or ER were seeded at a density of 10,000 cells/well in a 12-well plate. Cells were treated with 4-hydroxytamoxifen (4-OHT, 100 nM, SigmaAldrich, St. Louis, MO, USA) or vehicle (ethanol) 24 h after seeding (day 0). Media was changed, and a fresh dose of either 4-OHT or vehicle was added every 48 h. Cells were fixed every 24 h during 6 days by first washing twice with phosphate-buffered saline (PBS) and then incubating 10 min at room temperature with a 1% glutaraldehyde solution (in PBS). Cells were further washed twice with PBS and when all time points had been collected, they were stained with a crystal violet solution (0.3% in PBS) for 30 min under agitation. Crystal violet in excess was removed by immersion washes in water 10 times. Once the plates were dry, the fixed crystal violet cell staining was dissolved with an acetic acid solution (10% in distilled water) under agitation for 30 min, and optical density was measured at 590 nm to evaluate relative growth.

### 2.6. Western Blots

Cells and extracellular vesicles were collected in a Laemmli buffer containing Tris-HCl pH 8 (120 mM), glycerol (20% *v*/*v*), and sodium dodecyl sulfate (4% *m*/*v*). Total proteins were quantified with NanoDrop (absorbance at 280 nm). For kinetics of ∆Raf-1-ER activation, 20 µg of protein extracts was loaded on bilayered SDS-PAGE with 15% acrylamide in the lower half and 8.5% acrylamide in the upper half of the gel under a 4% stacking layer. BLUeye Prestained Protein Ladder (FroggaBio, Concord, ON, USA) was used as a molecular ladder. Proteins were transferred to the nitrocellulose membrane (Biorad, Saint-Laurent, QC, Canada). Primary antibodies used for immunoblotting were anti-estrogen receptor alpha (1:1000, clone F-10, sc-8002, Santa Cruz Biotechnology, Dallas, TX, USA), anti-p44-42 MAPK (Erk1/2) (1:1000, #9102, Cell Signaling Technology, Danvers, MA, USA), anti-Phospho-p44/42 MAPK Erk1/2 (Thr202/Tyr204) (1:1000, clone D13.14.4E, #4370, Cell Signaling Technology), anti-Histone H3 (1:2000, ab1791, Abcam, Cambridge, UK), anti-Phospho-histone H3 (Ser10) (1:1000, #06-570, Millipore), anti-α-Tubulin (1:10,000, clone B-5-1-2, T6074, Sigma). Secondary antibodies coupled to peroxidase (Biorad) and ECL detection reagent were used to reveal the signals.

### 2.7. Detection of Senescence-Associated-β-Galactosidase Activity

Senescence-associated-β-Galactosidase (SA-β-Gal) staining was performed 72 h post 4-OHT-induction in RAW 264.7 macrophages expressing ∆Raf-1:ER. Cells were washed twice with PBS at pH 5.5 and fixed for 15 min at room temperature with a 0.5% glutaraldehyde solution (in PBS pH 5.5). Fixed cells were washed twice with a MgCl_2_ solution (1 mM MgCl_2_ in PBS pH 5.5). Cells were then incubated at 4 °C overnight in a 0.2 µm filtered X-Gal solution (1 mg/mL 5-bromo-4-chloro-3-indolyl-beta-D-galactopyranoside, 5 mM K_3_Fe(CN)_6_, 5 mM K_4_Fe(CN)_6_∙3H_2_O, in PBS MgCl_2_ at pH 5.5). Cells were further incubated at 37 °C until the development of blue coloration (approximately 2–6 h). The percentage of blue cells was calculated by counting the number of blue cells from 100 cells in 3 different areas.

### 2.8. RNA Extraction, cDNA Synthesis, and QPCR

Cells and extracellular vesicles were collected in 1 mL TRIzol (Invitrogen) to isolate total RNAs. RNAs were extracted by phase separation by adding 200 µL of chloroform to 1 mL samples in TRIzol and centrifuging at 12,000× *g* for 15 min at 4 °C. The upper aqueous phase was collected and incubated for 10 min at room temperature with 500 µL of isopropanol. RNAs were precipitated by centrifuging at 12,000× *g* for 10 min at 4 °C. The RNA pellet was washed with 1 mL of 75% ethanol and centrifuged at 12,000× *g* for 5 min at 4 °C. The RNA pellet was left to air dry before resuspension in RNAse-free water at 55 °C for 20 min. The concentration and purity of total RNAs were measured with NanoDrop using absorbance at 260 nm vs. 280 nm and 230 nm.

To study messenger RNAs, reverse transcription of mRNAs was performed on 2 µg of total RNAs with 4 µL of 5X All-In-One RT MasterMix (Applied Biological Materials, Richmond, BC, Canada) in a final volume of 20 µL (incubations: 25 °C, 5 min; 37 °C, 5 min; 42 °C, 60 min; 85 °C, 5 min). The resulting cDNA was diluted 10 times with water and stored at −20 °C. Reactions for QPCR were performed in technical triplicate using 1 μL of diluted cDNA samples per 10 μL reaction volume, also containing the following: 0.25 µM of each primer (Biocorp, Pierrefonds, QC, Canada), 0.2 mM dNTP (DD0056, BioBasic, Markham, ON, USA), 0.33X Syber Green I (S7563, Invitrogen, Carlsbad, CA, USA), 0.25 U Jump Start Taq DNA polymerase (D9307, MilliporeSigma, Burlington, MA, USA) in 1X reaction buffer (provided with the enzyme), enriched with an additional 2.5 mM MgCl_2_ (M1028, Sigma). The LightCycler 96 real-time PCR system (Roche Applied Science, Penzberg, Germany) was used to detect the amplification level and was programmed to an initial step of six minutes at 95 °C, followed by 50 cycles of 20 s at 95 °C, 20 s at 58 °C, and 20 s at 72 °C. A high-resolution melting from 60 °C to 98 °C followed the amplifications. All reactions were run in triplicate, and the average values were used for relative quantification of target genes using the ∆∆CT method. Relative mRNA expression was normalized over the expression of two housekeeping genes: HMBS and TBP in human samples and β-Actin and Tbp in murine samples.

To study microRNAs, a polyadenylation step was performed before reverse transcription. Polyadenylation was performed on 2 µg of total RNAs with E. coli Poly(A) Polymerase and its buffer (NEB), in a final volume of 10 µL. (incubations: 37 °C, 50 min; 65 °C, 20 min). The resulting poly-adenylated RNA was reverse transcribed using OneScript Plus cDNA Synthesis kit (ABM) and a universal reverse primer [54] containing the sequence for both a TaqMan probe and a reverse QPCR primer (incubations: 70 °C, 5 min; 25 °C, 5 min; 37 °C, 5 min; 42 °C, 60 min; 85 °C, 5 min). The resulting cDNA was diluted 10 times with water and stored at −20 °C. Again, QPCR reactions were performed in technical triplicate using 0.33 μL of diluted cDNA samples per 10 μL reaction volume, this time containing: 1.5 µM of each specific forward and universal reverse primers (Biocorp), 0.2 mM dNTP (DD0056, BioBasic), 50 nM TaqMan Universal probe (IDT), 0.25 U Jump Start Taq DNA polymerase (D9307, MilliporeSigma) in 1X reaction buffer (provided with the enzyme), enriched with an additional 2.5 mM MgCl_2_ (M1028, Sigma). The LightCycler 96 real-time PCR system (Roche Applied Science) was used to detect the amplification level and was programmed to an initial six-minute step at 95 °C, followed by 45 cycles of 15 s at 95 °C and 60 s at 60 °C. All reactions were run in triplicate, and the average values were used for relative quantification of target genes using the ∆∆CT method. Relative miRNA expression was normalized over the expression of two housekeeping genes: snRNA U6 and rRNA 5S. Primers are described in Table 1 and Table 2.

### 2.9. Immunofluorescence

RAW 264.7 cells expressing ∆Raf-1:ER were seeded on a coverslip in 6-well plates at a density of 250,000 cells/well and treated with 4-OHT (100 nM) or at a density of 125,000 cells/well and treated with vehicle (EtOH). A second treatment was performed 48 h after the first one. After 72 h in culture on the coverslip, cells were rinsed once with PBS and fixed in a solution of 4% paraformaldehyde in PBS for 10 min at room temperature (RT). Two more washes in PBS followed, and fixed cells were then kept at 4 °C in PBS + 0.02% Azide until immunostaining. Cells were washed twice for 5 min at RT with PBS with 0.1 M Glycin to remove azide and then permeabilized with 0.1 M Glycine and 0.4% Triton X100 in PBS for 5 min at 4 °C. Then, they were incubated 3 times for 15 min with 3% BSA in PBS. The primary antibody was diluted in 3% BSA in PBS. Primary antibody was added to the coverslip and incubated in a humidified chamber at 4 °C overnight. The next day, the cells were washed 3 times with PBS 3% BSA for 10 min. Species-specific Alexa-fluor conjugated secondary antibody (Life Technologies, Carlsbad, CA, USA) was diluted 1:1500 in 3% BSA in PBS and incubated for 1 h at RT. Cells were then washed three times with PBS, excess PBS was removed, and coverslips were mounted on glass slides with Vectashield containing DAPI. Edges were sealed off with nail polish and mounted cells were kept for a minimum of 24 h at 4 °C. On the day of confocal imaging, coverslips were removed from the fridge at least 1 h before imaging and put in a microscope box to warm up to room temperature. For further confocal microscope imaging, the Zeiss LSM 800 with a spectral analysis detector was used. All images were acquired sequentially. The data were acquired with a maximal airy unit and processed using Zen Blue V26 software and ImageJ 1.53t.

Primary antibodies used for immunofluorescence were phospho-Ser 139 H2A.X (1:100, clone JBW-301, cat #05-636 Millipore), 53BP-1 (1:200, clone Ab-1, at #PC712 Calbiochem, San Diego, CA, USA), Lamin B1 (1:450, cat #ab16048 Abcam, Cambridge, UK), PML (1:300, produced by our laboratory against the peptide comprising amino acids 352–356 of human PML-IV, a region common to all PML isoforms) [55]. This sequence is present in all known PML isoforms. The peptide was coupled with KLH and used to immunize two rabbits. Both rabbits developed anti-PML immunoreactivity (MF1 and MF2), as assessed in Western blots using extracts of cells from MEFs wild-type or KO for PML (Appendix A) and senescent human fibroblasts (Appendix A).

Lung samples from Kras and ATTAC-Kras mice were described in [27]. They were treated with 2 mg/kg of body weight AP20187 (AP; B/B homodimerizer; Clontech/Takara, Kusatsu, Shiga**,** Japan) twice a week, beginning within the first week of birth. Lung adenomas and flanking ‘normal’ lung tissue were micro-dissected after 8 weeks, fixed with 4% paraformaldehyde (PFA) for two hours at 4 °C, and then transferred to a 30% sucrose solution for 24 h, at 4 °C. Samples were O.C.T.-embedded (Tissue-Tek #4583, Sakura Finetek USA, Torrance, CA, USA) and stored at −80 °C. For this study, they were retrieved from the freezer, permeabilized with 0.5% Triton in 1X PBS for 20 min at RT, and incubated in blocking solution (5% BSA in 1X PBS) for 30 min at RT. Incubation with primary antibodies PLAUR (R&D #AF534; 1:100) and F4/80 (Cell Signaling #71299; 1:250) was performed overnight at 4 °C, followed by 1X PBS washes and incubation with secondary antibodies at RT for 2 h. Lastly, samples were stained with DAPI and stored at 4 °C until imaged. Imaging was performed using the Zeiss LSM 780 confocal system.

### 2.10. RNAseq

RNAs from cells collected in TRIzol were extracted by phase separation with chloroform, as described above. A total of 600 µL of the upper aqueous phase was collected and incubated for 10 min at room temperature (RT) with 300 µL isopropanol. This solution was loaded onto a RNeasy column (Qiagen Mini Kit #74106, Hilden, Germany) and centrifuged at 12,000 rpm for 30 s at RT. The column was centrifuged again at 12,000 rpm for 30 s at RT after adding 700 µL of RW1 buffer to wash away non-RNA biomolecules. Residual salts were removed by washing twice with 500 µL of RPE buffer and centrifuging at 12,000 rpm for 30 s at RT. The column was dried by centrifuging at 12,000 rpm for 2 min at RT. A total of 50 µL of RNAse-free water was incubated for 2 min on the column before elution of RNAs by centrifuging at 12,000 rpm for 1 min at RT. RNAs were quantified with Qubit BR. A total of 1000 ng of total RNA was used for library preparation. The quality of total RNA was assessed with the BioAnalyzer Nano (Agilent, Santa Clara, CA, USA), and all samples had an RIN above 9.5. Library preparation was performed with the KAPA mRNAseq Hyperprep kit (KAPA, Cat no. KK8581, Roche, Laval, QC, USA). Ligation was made with 64 nM final concentration of Illumina index and 9 PCR cycles were required to amplify cDNA libraries. Libraries were quantified by QuBit and BioAnalyzer DNA1000. All libraries were diluted to 10 nM and normalized by QPCR using the KAPA library quantification kit (KAPA; Cat no. KK4973). Libraries were pooled to equimolar concentration. Sequencing was performed with the Illumina Nextseq500 using the Nextseq High Output Kit (86 cycles) using 2.2 pM of the pooled libraries. Around 20 M single-end PF reads were generated per sample. Library preparation and sequencing were performed at the Genomics Platform of the Institute for Research in Immunology and Cancer (IRIC).

### 2.11. Analysis of RNA-Seq Data

The initial reads were first cleaned by removing adapter sequences used for sequencing and low-quality sequences using Trimmomatic version 0.35 [56]. Reads were then aligned to the mouse genome GRCm38 using STAR version 2.5.1b [57]. Read counts obtained from STAR were used as a measure of gene expression, and they were computed using RSEM [58] to obtain gene and transcript-level expression, expressed in either TPM (Transcript per Million) or FPKM (The Fragments per Kilobase Million) values, for these stranded RNA libraries. DESeq2 version 1.18.1 [59] was then used to normalize gene read counts. Data are available at GEO: GSE153508. GSAE (Gene Set Enrichment Analysis, version 4.0.3) [60] and DAVID (Database for Annotation, Visualization and Integrated Discovery, https://david.ncifcrf.gov/tools.jsp) [61] were used to identify gene expression patterns and functional annotation of differentially expressed RNAs from senescent macrophages and senescent EVs. DiRE (distant regulatory elements of co-regulated genes, https://dire.dcode.org/) was used to predict transcription factors that bind differentially expressed genes. Heatmaps for differentially expressed genes were prepared using GraphPad Prism.

### 2.12. Extracellular Vesicles

Senescent and non-senescent macrophages were cultured in serum-free culture media 48 h before extracellular vesicles harvest. Extracellular vesicles (EVs) were separated from the culture medium by polymer-based precipitation using ExoQuick-TC (SBI—System Biosciences, Palo Alto, CA, USA) according to the manufacturer’s protocol. In short, the culture medium was collected and centrifuged at 3000× *g* for 15 min to remove cellular debris. ExoQuick-TC reagent was added to the supernatant in a ratio of 1 mL to 5 mL of culture media. Samples were mixed by inversion and incubated at 4 °C overnight. EVs were collected by centrifuging samples at 1500× *g* for 30 min and the supernatant was removed. A second centrifugation was performed at 1500× *g* for 5 min to remove residual liquid. The EV pellet was resuspended either in TRIzol for RNA extraction or in Laemmli buffer for Western blots.

For fluorescent Nanoparticle Tracking Analysis (fNTA), small RNAseq analysis, and proteomics analysis, EVs were isolated at SBI from frozen conditioned media sent to them using ExoQuick-TC. The fNTA was performed by SBI by labeling EVs with the ExoGlow-NTA dye (SBI) to determine the abundance and size of EVs. For small RNAseq analysis, SBI isolated total RNA from EVs using the SeraMir Exosome RNA Purification Column kit (Cat #RA808A-1, SBI) according to their available protocols. For each sample, 1 µL of the final RNA eluate was used for measurement of small RNA concentration by Agilent Bioanalyzer Small RNA Assay using Bioanalyzer 2100 Expert instrument (Agilent Technologies, Santa Clara, CA, USA). Small RNA libraries were constructed with the CleanTag Small RNA Library Preparation Kit (TriLink, Cat# L-3206, San Diego, CA, USA) according to the manufacturer’s protocol. The final purified library was quantified with High Sensitivity DNA reagents (Agilent Technologies, PO# G2933-85004) and High Sensitivity DNA chips (Agilent Technologies, PO# 5067-4626). The libraries were pooled, and the 140 bp to 300 bp region was size-selected on an 8% TBE gel (Invitrogen by Life Technologies, Ref# EC6215). The size-selected library is quantified with High Sensitivity DNA 1000 Screen Tape (Agilent Technologies, PO# 5067-5584), High Sensitivity D1000 reagents (Agilent Technologies, PO# 5067-5585), and the TailorMix HT1 QPCR assay (SeqMatic, Cat# TM-505), followed by a NextSeq High Output single-end sequencing run at SR75 using NextSeq 500/550 High Output v2 kit (Cat# FC-404-2005, Illumina, San Diego, CA, USA) according to the manufacturer’s instructions. NGS Library generation and sequencing were performed by System Biosciences. Small RNAs from frozen cell pellets were similarly processed by SBI. Differentially expressed genes were identified using the DESeq package by first normalizing counts across samples and determining the variance among samples. The analyzed results were made available via the Maverix Analytic Platform.

### 2.13. Mass Spectrometry

After purification, EVs were incubated with 0.017% NaDOC for 30 min on ice and precipitated overnight at 4 °C in 6% TCA. After spinning for 60 min at max speed, the pellets were washed twice with ice-cold acetone and air-dried. The samples were then reduced in urea and DTT, alkylated in iodoacetamide, and then digested overnight at 37 °C, 1200 RPM with 260 ng Trypsin (Promega V5111). The reaction was quenched in a final concentration of 0.71% formic acid and 4.57 mM TCEP. Tryptic digests were dried.

Before LC-MS/MS, protein digests were re-solubilized under agitation for 15 min in 10 µL of 0.2% formic acid. Desalting/cleanup of the digests was performed by using C18 ZipTip pipette tips (Millipore, Billerica, MA, USA). Eluates were dried down in a vacuum centrifuge and then re-solubilized under agitation for 15 min in 11 µL of 2%ACN/1% formic acid. The LC column was a PicoFrit fused silica capillary column (15 cm × 75 µm i.d., New Objective, Woburn, MA, USA), self-packed with C-18 reverse-phase material (Jupiter 5 µm particles, 300 Å pore size; Phenomenex, Torrance, CA, USA) using a high-pressure packing cell. This column was installed on the Easy-nLC II system (Proxeon Biosystems, Odense, Denmark) and coupled to the Orbitrap Fusion mass spectrometer (Thermo Scientific) through a Nanospray Flex Ion Source. The buffers used for chromatography were 0.2% formic acid (buffer A) and 100% acetonitrile/0.2% formic acid (buffer B). Peptides were loaded onto the column at a flow rate of 600 nL/min and eluted with a 2-slope gradient at a flow rate of 250 nL/min. Solvent B first increased from 2 to 25% in 15 min and then from 25 to 80% B in 34 min.

Nanospray and S-lens voltages were set to 1.6 kV and 60 V, respectively. Capillary temperature was set to 250 °C. Full scan MS survey spectra (*m*/*z* 360–1560) in profile mode were acquired in the Orbitrap with a resolution of 120,000, a target value at 1 × 10^6^, and a maximum injection time of 50 ms. The 22 most intense peptide ions were fragmented in the HCD collision cell and analyzed in the linear ion trap with a target value at 2 × 10^4^, a maximum injection time of 50 ms, and normalized collision energy at 28 V. Target ions selected for fragmentation were dynamically excluded for 15 s after two MS2 events.

For protein identification, the peak list files were generated with Proteome Discoverer (version 2.1) using the following parameters: minimum mass set to 500 Da, maximum mass set to 6000 Da, no grouping of MS/MS spectra, precursor charge set to auto, and minimum number of fragment ions set to 5. Protein database searching was performed with Mascot 2.6 (Matrix Science, Boston, MA, USA) against the Refseq_Human database (February 2018). The mass tolerances for precursor and fragment ions were set to 10 ppm and 0.6 Da, respectively. Trypsin was used as the enzyme, allowing for up to 1 missed cleavage. Cysteine carbamidomethylation was specified as a fixed modification, and methionine oxidation as a variable modification. Data interpretation was performed using Scaffold (version 4.11.1).

### 2.14. Senescence Escape

RAW 264.7 cells expressing ∆Raf-1:ER were seeded in 10 cm plates at a density of 2 × 10^6^ cells and treated with 4-OHT (100 nM) or at a density of 1 × 10^6^ cells and treated with vehicle (EtOH) every two days. Culture media was replaced 24 h later with fresh media containing 5% FBS to remove 4-OHT and EtOH on RAW 264.7 cells. The medium conditioned for 24 h was then filtered (0.45 µm) and transferred to senescent MEFs initially seeded in a 6-well plate at a density of 35,000 cells/well. The transfer of conditioned media was made every day. After a total of 21 days of conditioned media transfer from RAW 264.7, senescent MEFs cultures were fixed in 1% glutaraldehyde. The number of colonies was assessed by a crystal violet retention assay as detailed above.

### 2.15. Senolytic Assay

RAW 264.7 cells expressing ∆Raf-1:ER were seeded at a density of 20,000 cells/well in a 12-well plate. Cells were treated with 4-OHT (100 nM) or vehicle (ethanol) 24 h after seeding. Media were changed and cells were treated with Navitoclax ABT-263 (10 µM) or vehicle (DMSO) 24 h after 4-OHT treatment. Cells were fixed in 1% glutaraldehyde 48 h following Navitoclax treatment. Viability was assessed by a crystal violet retention assay, as detailed above.

### 2.16. Statistical Analysis

We used Microsoft Excel and GraphPad Prism to perform statistical analysis and to plot the data as the mean ± standard deviation (SD). Unpaired Student’s *t* tests were used to compare the experimental group with the control. The Mann–Whitney U test was used to compare IF data. Values of *p* < 0.05 were considered statistically significant.

## 3. Results

### 3.1. Hyperactive ERK Signaling Induces Senescence in Murine Macrophages

To establish an in vitro model of senescence in macrophages (MΦ), we used a retroviral vector that expresses a conditional allele of the oncoprotein Raf-1 called ∆RAF-1:ER in RAW 264.7 mouse MΦ. This fusion protein consists of the kinase domain of Raf-1 fused with the ligand-binding domain of the estrogen receptor alpha (ER). Treatment with 4-hydroxytamoxifen (4-OHT), a ligand of ER, allows the release of Raf-1 activity and further signaling activation. This regulated system is used to induce oncogene-induced senescence by hyperactivating the mitogen-activated protein kinases (MAPK) pathway. Of note, hyperactive ERK characterizes senescence in cell culture in response to short telomeres [62,63] and in vivo both during premature aging [64] or natural aging [65,66,67,68], making this model relevant to the accumulation of senescent cells seen in old organisms.

We first validated the induction of senescence in our model by a panel of senescence markers. In RAW 264.7 macrophages, Raf-1 activation leads to growth arrest (Figure 1A). This effect is not directly caused by 4-OHT, as RAW 264.7 cells expressing ER do not exhibit this growth arrest under 4-OHT. By Western blot, we confirmed that ∆Raf-1:ER is expressed in RAW 264.7 cells (Figure 1B). Treatment with 4-OHT initially increases ∆Raf-1:ER expression, while the expression decreases after 4 days of treatment. As expected, treatment with 4-OHT increases the phosphorylation of Erk1/2 by activating the MAPK pathway (Figure 1B). This activation is, however, transient and is lost after about 4 days of 4-OHT treatment, in agreement with ∆Raf-1:ER level reduction at the same time points. Moreover, Raf-1 activation is associated with a decrease in the levels of the mitosis marker phosphorylated Histone H3 (Figure 1B). We next investigated the levels of transcripts used as senescence markers. Cdkn1a (p21), Pml, Fam214b, Tgfb1, Pai-1, Angptl2 are significantly upregulated following treatment with 4-OHT, while Ki67, C2cd5, Cntln, Patz1, Trdmt1 are downregulated by Raf-1 activation. (Figure 1C,D). We also observed an accumulation of cells with elevated senescence-associated-β-Galactosidase (SA β-Gal) activity and cells displaying an enlarged morphology (Figure 1E). Raf-1 activation also increased the number of cells with Lamin B1 defects (Figure 1F,G). Furthermore, the presence of 53BP-1 and γ-H2A.X foci suggested an activation of the DNA damage response in RAW 264.7 macrophages following treatment with 4-OHT (Figure 1H,I). An elevation in the quantity of nuclear PML foci was also observed in RAW 264.7 cells with activated Raf-1 (Appendix A). Finally, the senolytic agent Navitoclax (ABT-263) reduced the number of senescent MΦ but had little effect on the number of control MΦ (Appendix A). Together, these results suggest that activation of Raf-1 induces cellular senescence in RAW 264.7 MΦ by hyperactivating the MAPK pathway.

### 3.2. Enrichment of Inflammatory Genes in the Transcriptome of Senescent Macrophages

To gain insight into the gene expression of senescent macrophages, we compared the transcriptome of control (Ctrl) and senescent (Sen) macrophages (MΦ) by RNA sequencing (RNAseq). A volcano plot representation of these data highlights 1446 upregulated genes and 304 downregulated transcripts in Sen MΦ, compared with Ctrl MΦ (Figure 2A). Some of the most highly enriched transcripts in Sen MΦ include Itga3, Itgb3, Gm5483, Upp1, Slc9a2, Tm4sf1, Tmem132b, and Mcpt1. The receptor for the urokinase plasminogen activator Plaur, the target of senolytic CAR T cells [69], was also upregulated in these senescent MΦ (GEO: GSE153508). Distant regulatory elements (DiRE) algorithm predicted NF-κB to be the most important transcription factor to explain the co-regulation of the top 100 most enriched transcripts in Sen MΦ (Figure 2B). To further investigate the expression signature of senescent macrophages, we performed Gene Set Enrichment Analysis (GSEA) with our RNAseq data (Figure 2C). We determined that genes found in *HALLMARK MYC TARGETS V1* and *HALLMARK E2F TARGETS* gene sets were downregulated in Sen MΦ consistent with a senescent cell-cycle arrest. Furthermore, there was an upregulation of genes from the sets *HALLMARK INFLAMMATORY RESPONSE* and *HALLMARK TNFA SIGNALING* VIA *NFKB*, suggesting that Sen MΦ have an important pro-inflammatory SASP signature. Indeed, RNAseq revealed a wide range of cytokines upregulated in Sen MΦ (Figure 2D). Curiously, classic SASP markers, such as IL6 or IL8, were not identified as being upregulated in Sen MΦ. To validate some of our RNAseq findings, we performed RT-QPCR on Ctrl and Sen MΦ (Figure 2E). We found that Gm5483, Itga3, Itgb3, Mcpt1, Slc9a2, Tm4sf1, Tmem132b, and Upp1 were highly upregulated in Sen MΦ, reproducing the data from the RNAseq. Together, these results suggest that inflammation is a strong hallmark of senescent macrophages, and that NF-κB could regulate the expression of this program. Moreover, the composition of the SASP of Sen MΦ might be distinct from the one from other senescent cell types.

To confirm the fact that some of the biomarkers found in the RAW 264.7 cell model also occur in senescent macrophages induced by other methods, we first isolated bone marrow-derived macrophages (BMDM) from the p16-3MR mouse model [70]. These animals were engineered to express three genes under the control of the p16Ink4a promoter: a renilla luciferase, a monomeric red fluorescent protein, and the herpes simplex virus 1 thymidine kinase. After isolation, the BMDM were treated with macrophage colony stimulating factor (M-CSF) to induce their differentiation and 500 μM of H_2_O_2_ to induce senescence. Most cells treated with H_2_O_2_ acquired a red fluorescence (Appendix A), indicating activation of the p16INK4a promoter. Additionally, they expressed high levels of Cdkn1a (p21) and Plau (urokinase), as in senescent RAW 264.7 macrophages. However, the levels of Tm4sf1, SerpinE1, and Plaur were not significantly upregulated, although there appeared to be a trend towards higher values (Appendix A). In contrast, we measured the levels of Plaur in lungs bearing KRAS-driven tumors, previously shown to accumulate p16Ink4a+ macrophages with a gene expression pattern characteristic of senescent cells [27]. We found that Plaur increased in tumor-bearing lungs in comparison to normal lungs (Appendix A). Notably, this increase was reduced in INK-ATTAC mice, where senescent cells were eliminated after treatment with AP20187 (AP), which activates a conditional caspase-8 allele expressed under the control of the p16INK4a promoter (Appendix A) [27]. These results indicate both common and distinctive gene expression biomarkers in different models of senescent MΦ.

### 3.3. Senescent Macrophages Secrete More EVs Carrying RNA-Binding Proteins and Exosome Proteins

As growing evidence suggests that EVs are highly secreted by senescent cells, we investigated the EVs of Sen MΦ. Fluorescent Nanoparticle Tracking Analysis (fNTA) revealed that Sen MΦ secrete more EVs than Ctrl MΦ, although the size of EVs is similar in both conditions (Figure 3A). The proteome of EVs secreted by Ctrl and Sen MΦ was analyzed by Mass Spectrometry (MS). A total of 124 proteins were identified in all EVs, with a minimum of 10 exclusive spectrum counts. A gene ontology (GO) analysis of these proteins highlighted GO terms referring to EVs and exosomes. Of these 124 proteins, 29 were enriched, and 2 were depleted in EVs from Sen MΦ compared to those of Ctrl MΦ (Appendix A and Figure 3B). As expected, some proteins enriched in EVs from Sen MΦ were associated with the GO term vesicle. However, more than half of the enriched proteins in Sen MΦ EVs were labeled as intracellular and/or ribosomal subunits, suggesting a senescence-specific targeting of some proteins to EVs (Figure 3B). Moreover, many proteins enriched in the EVs of Sen MΦ are positive regulators of assembly and secretion of exosomes, suggesting they could be responsible for the abundant EV secretion of Sen MΦ (Appendix A). We validated by immunoblots the increase of both Cd9 and Mvp in EVs from Sen MΦ. The endoplasmic reticulum marker Grp94 and BAS were absent in the EVs preparation, showing the lack of cellular or media contaminants (Figure 3C).

### 3.4. EVs from Senescent Macrophages Carry Pro-Inflammatory miRNAs

To investigate the RNA cargo of Sen MΦ EVs, a small RNAseq analysis was performed. The RNAs of both Ctrl and Sen MΦ and their derived EVs were compared. First, we compared the types of small RNAs regulated in each condition (Figure 4A). The majority of enriched RNAs in EVs from Sen MΦ compared with EVs from Ctrl MΦ are mRNAs, antisense transcripts, and miRNAs. Interestingly, piRNAs are largely depleted from the EVs of Sen MΦ, although they are not downregulated in their producing cells. This could suggest the existence of a mechanism to exclude piRNAs from EVs in senescence. We next focused our analysis on the miRNA cargo of EVs from Sen MΦ, since they have a potential to regulate gene expression in recipient cells. In Sen MΦ, in comparison to Ctrl MΦ, there were 64 miRNAs identified as downregulated, which is almost twice more miRNAs than those upregulated. However, in Sen MΦ EVs compared with Ctrl MΦ EVs, there were 95 miRNAs identified as enriched, which is roughly five times more than the number of depleted miRNAs (Figure 4B). This means that the EVs of Sen MΦ carry a wide variety of miRNAs that do not reflect the miRNA expression change at the intracellular levels, suggesting a regulated incorporation of miRNAs inside EVs in senescence. Moreover, when comparing the number of reads mapped to miRNAs or other RNAs, we found that miRNAs represented 25% of the total reads in the EVs of Sen MΦ, which is twice as much as in all other conditions (Figure 4C). This suggests that miRNAs represent a large portion of the RNA cargo in EVs from Sen MΦ. Furthermore, this fraction goes up to almost 50% when considering only regulated RNAs in EVs from Sen MΦ compared with EVs from Ctrl MΦ (Figure 4C). This could mean that senescence promotes the incorporation of miRNAs in EVs. Some of the most enriched miRNAs in the EVs of Sen MΦ were the pro-inflammatory miRNAs miR-132-3p, miR-21a-5p, and miR-155-5p, suggesting that EVs share the ability to induce inflammation with the soluble factors of the SASP. QPCR confirmed the enrichment of miR-132-3p, miR-21a-5p, and miR-155-5p in the EVs of Sen MΦ (Figure 4D). Overall, small RNAseq findings indicate that miRNAs are abundant in the EVs of Sen MΦ, and some of these miRNAs have known pro-inflammatory functions that could affect recipient cells.

Although the procedure for small RNAseq targets preferentially miRNAs, we did detect several mRNA fragments in EVs. We do not know whether these are degradation products or represent functional mRNAs. However, there were more upregulated mRNAs when we compared the EVs from Sen MΦ with the EVs from Ctrl MΦ (Appendix A). In contrast, inside the senescent cells, the downregulated mRNA population was more important (Appendix A). The number of regulated mRNAs reads was higher in EVs from Sen MΦ vs. EVs from Ctrl MΦ (Appendix A). A gene ontology analysis of the upregulated mRNAs from the EVs of Sen MΦ revealed functions in apoptosis, antiviral defenses, angiogenesis, aging, inflammation, the MAP kinase pathway, and lipid metabolism (Appendix A). Hence, the EVs from Sen MΦ, in general, have more RNA, including mRNAs, miRNAs, and perhaps other RNA species, but not piRNAs.

### 3.5. Secretions of Senescent Macrophages Promote Escape from Senescence in MEFs

To investigate the potential effects of Sen MΦ on other cells, we did a conditioned media transfer from either Ctrl or Sen MΦ during 21 days onto already senescent mouse embryonic fibroblasts (MEFs, Figure 5A,B). We observed by crystal violet retention assay the formation of more colonies in the MEF population cultured with the conditioned media of Sen MΦ in comparison with MEFs cultured with either regular media or conditioned media of Ctrl MΦ (Figure 5C). These results suggest that the secretions of Sen MΦ promote the escape from senescence in MEFs. This effect could be the result of SASP soluble factors and/or EVs carrying pro-inflammatory and pro-proliferative miRNAs. Interestingly, the EVs secreted by Sen MΦ have the potential to induce paracrine proliferation. In fact, miR-21 can induce proliferation by targeting Pdcd4 and Pten (87–88), while miR-155 has been shown to promote proliferation by targeting Tp53inp1, which decreases the expression of p53 (89). In addition, miR-155 decreased the expression of p21 by targeting Smad5 (90). Moreover, SASP soluble factors also have the potential to induce paracrine proliferation. In fact, the fibroblast growth factor Fgf2, also increased in Sen MΦ, can induce an escape from senescence in human mesenchymal stem cells by decreasing the expression of TGF-β2, p53, p21 et p16 (91). Therefore, SASP soluble factors and miRNAs from secreted EVs probably work in synergy to induce proliferation in senescent MEFs.

## 4. Discussion

Here, we characterized the senescence response to aberrant activation of ERK signaling in the murine macrophage cell line RAW 264.7, which has been used extensively to characterize the molecular mechanisms and gene expression regulation relevant to macrophage biology [71]. The senescence response of these cells includes the expression of several biomarkers previously identified in other cell types, including growth arrest, increases in SA-β-Gal, DNA damage, and PML bodies, as well as a reduction in cell proliferation markers and Lamin B1. In addition, our RNAseq data indicated an upregulation of more than 50 cytokines in senescent RAW 264.7 macrophages. Intriguingly, classic SASP markers Il-6 and Il-8 were not upregulated in these cells. A recent study aiming to identify ubiquitous members of the SASP throughout different senescence models also observed that upregulation of Il-6 and Il-8 was not shared between all these models [36]. Moreover, our senescent MΦ exhibit an upregulation of Stc1, which was identified as a ubiquitous SASP marker by the same study. Senescent MΦ also shared a part of the SASP signature of Ras-induced senescence, with inductions of Cxc110, Mmp3, and Mmp9 [36]. The SASP of senescent MΦ could be under the transcriptional control of NF-κB, as suggested by our analysis with the algorithm DiRE. Moreover, our study revealed new potential senescent biomarkers for macrophages such as Slc9a2, Gm5483, Tm4sf1, and the chemokine receptors Cxcr3 and Ccr5.

As reported before [30,31,32,33,34,35], we found that the secretions of EVs were higher in senescent cells. Interestingly, Alix (also named Pdcd6ip), which is important for the generation of EVs [72], was found to be enriched in the EVs of senescent macrophages. In addition, the cargo of EVs from senescent macrophages contained increased levels of the miRNAs miR-21, miR-155, miR-132, ribosomal proteins, and the major vault protein (Mvp). Of interest, Mvp is upregulated in senescent cells, and its transcription is controlled by p53, a key player in senescence [73,74]. Mvp has previously been found to carry miR-193a to exosomes [75]. On the other hand, classic EV markers, such as tetraspanins Cd63 and Cd81, were not detected in EVs from both non-senescent and senescent macrophages, suggesting that these tetraspanins are not EV markers in macrophages. A recent study also found that tetraspanins were not ubiquitous EV markers among different cell lines [43]. Instead, the authors identified Syntenin-1 (also called Sdcbp) as a broadly conserved EV marker. Indeed, Syntenin-1 was detected in EVs from our senescent and non-senescent macrophages.

We observed that senescent EVs generally contain more miRNAs and mRNAs than non-senescent EVs. Further investigations are needed to determine the functional significance of this RNA cargo and the interactions between miRNAs and mRNAs delivered by the same EVs. Previous studies have confirmed the delivery of miRNAs via EVs [44,76,77,78]. However, delivering functional mRNAs capable of being translated in recipient cells is challenging due to rapid mRNA degradation [79]. Nevertheless, tumor cells can transfer functional mRNA cargo to recipient cells [80]. The combination of mRNAs and miRNAs in EVs may represent a broader signaling strategy. For example, it could facilitate the enhanced translation of incoming mRNAs from EVs while suppressing the translation of specific mRNAs in host cells. Many studies attribute EVs function to individual molecules, but they transport a cargo of bioactive molecules that may cooperate to regulate specific functions.

Macrophages influence the microenvironment in many tissues via a great variety of secretory products. This effect is commonly transient and highly regulated. However, when macrophages enter senescence, the constant efflux of pro-inflammatory products could initiate or reinforce a pathology. One of the processes that senescent macrophages can stimulate is malignant transformation [27,28], and the mechanisms behind this effect are just beginning to unravel. Our finding, that secretions from senescent macrophages can induce an escape from senescence, introduces a fresh perspective on the tumorigenic impact of these cells within a comprehensively understood mechanistic framework. This discovery opens the possibility of actively searching for secreted factors that can inhibit senescence, guided by the insights derived from these results.

## 5. Limitations of the Study

Although we identified an anti-senescence function of the senescent cell secretome, we could not define the key components necessary and sufficient for this effect. In our experimental setup, we needed to change the conditioned media from macrophages every day during the 21 days of culturing required to observe the formation of colonies by senescent cells, perhaps suggesting that the key factors are labile. In subsequent studies, we will need to modify and optimize the current assay so that genetic experiments can be performed to identify the specific anti-senescence factors. We can only speculate that there are factors both among the cytokines and the RNAs in EVs that can bypass senescence. They include miR-21a [81], miR-155 [82], and Fgf2 [83]. It will also be important to investigate whether this anti-senescence function of macrophages has any physiological role.

## Figures and Tables

**Figure 1 biomedicines-12-01089-f001:**
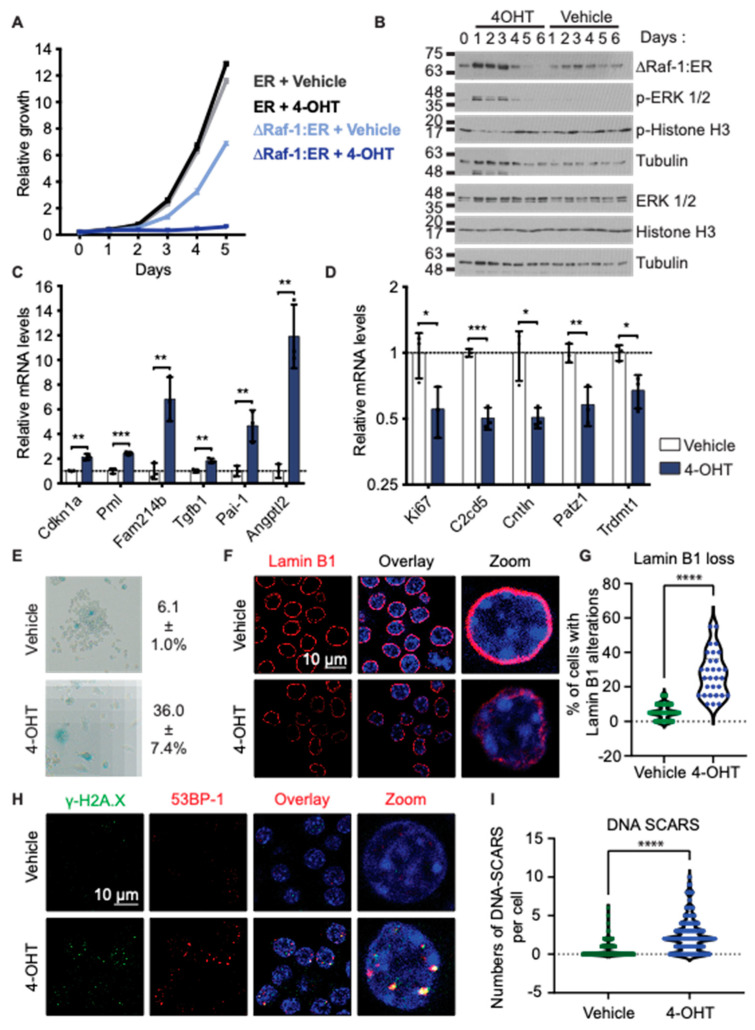
Raf hyperactivation induces cellular senescence in RAW 264.7 murine macrophages. (**A**) Growth curve of RAW 264.7 cells expressing ∆Raf-1:ER or ER treated every 48 h with vehicle (ethanol) or 4-hydroxytamoxifen (4-OHT, 100 nM). Relative growth was assessed by crystal violet retention assay. Each point represents the mean of a technical triplicate and error bars correspond to the standard deviation. The experiment was performed 3 times (the graph corresponds to one representative biological replicate). (**B**) Western blots showing levels of ∆Raf-1:ER (with ER-α antibody), phospho-ERK, and phospho-histone H3. Protein extracts were obtained from RAW 264.7 cells expressing ∆Raf-1:ER treated as in (A). Tubulin was used as a loading control. The experiment was performed 3 times (the figure corresponds to one representative biological replicate). (**C**,**D**) RT-QPCR of markers that are upregulated in senescence (**C**) and downregulated in senescence (**D**). RT-QPCR was performed on total RNA extract from RAW 264.7 cells expressing ∆Raf-1:ER and treated for 3 days with 4-OHT or vehicle, as in (**A**). RNA levels were normalized over Tbp and β-Actin. The experiment was performed 3 times. Error bars represent standard deviation; * *p*-value < 0.05; ** *p*-value < 0.01; *** *p*-value < 0.001 using Student’s *t*-test. (**E**) Senescence-associated ß-Galactosidase staining of RAW 264.7 cells expressing ∆Raf-1:ER treated as in (**C**,**D**). (**F**) Immunofluorescence (IF) staining of Lamin B1 with DAPI staining performed on RAW 264.7 cells, as in (**C**,**D**). (**G**) Quantification of IF from F showing the percentage of cells with Lamin B1 alterations. (**H**) Immunofluorescence (IF) staining of γ-H2AX, 53BP-1, and DAPI staining performed on RAW 264.7 cells expressing ∆Raf-1:ER and treated as in (**C**,**D**). (**I**) Quantification of IF from H showing number of cells with DNA segments with chromatin alterations reinforcing senescence (DNA-SCARS). For all IF, experiments were performed 3 times; **** *p*-value < 0.0001 using Mann–Whitney U test.

**Figure 2 biomedicines-12-01089-f002:**
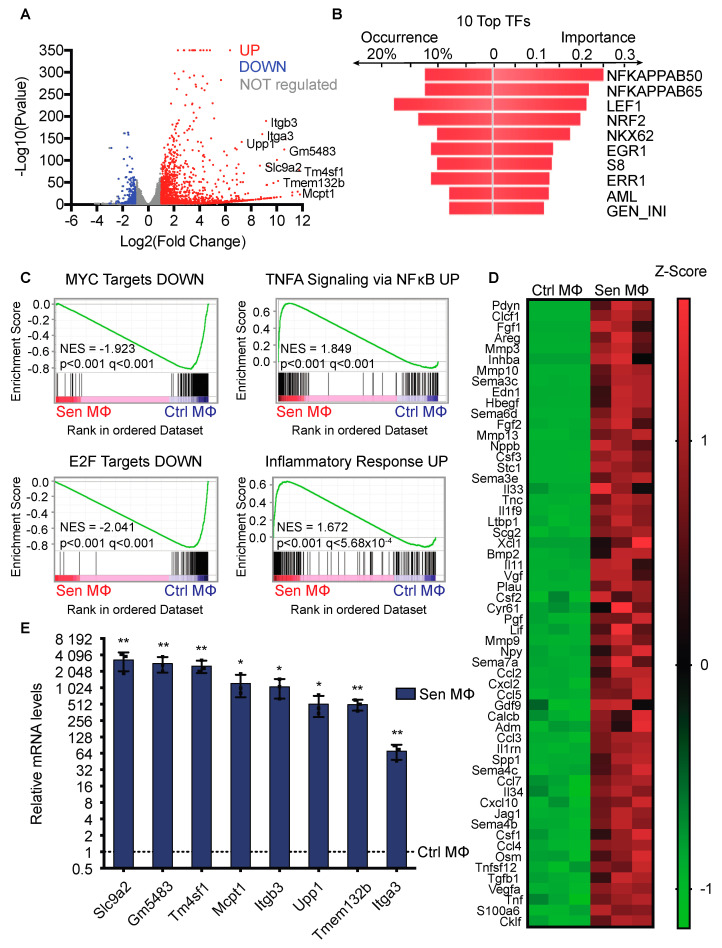
Enrichment of inflammatory genes in the transcriptome of senescent macrophages. (**A**) Volcano plot of the RNA sequencing (RNAseq) results comparing senescent and control macrophages. RNAseq was performed on 3 independent samples of both control and senescent macrophages. Upregulated and downregulated genes in senescent macrophages are reported as red dots and blue dots, respectively. Gray dots represent unregulated genes. Gene expression was considered regulated when its transcript levels changed two-fold, with a *p*-value ≤ 0.01 (*p*-adj-value ≤ 0.25). (**B**) DiRE analysis obtained from top 95 most significantly upregulated genes in senescent macrophages according to RNAseq results, showing the 10 most important transcription factors (DiRE: distant regulatory elements of co-regulated genes). Occurrence refers to the fraction of regulatory elements containing the binding site for the indicated transcription factor. (**C**) Hallmark signatures found by analyzing the regulated genes from RNAseq results with the Gene Set Enrichment Analysis (GSEA) software version MSigDB 2023.2. Sen MΦ: senescent macrophages, Ctrl MΦ: control macrophages, NES: normalized enrichment score. (**D**) Heatmap indicating upregulated cytokines in Sen MΦ from RNAseq results. (**E**) QPCR validation of upregulated transcripts in Sen MΦ found in A. The dotted line represents RNA levels of Ctrl MΦ. RNA levels were normalized over Tbp and β-Actin. The experiment was performed 3 times. Error bars represent standard deviation; * *p*-value < 0.05; ** *p*-value < 0.01 using Student’s *t*-test.

**Figure 3 biomedicines-12-01089-f003:**
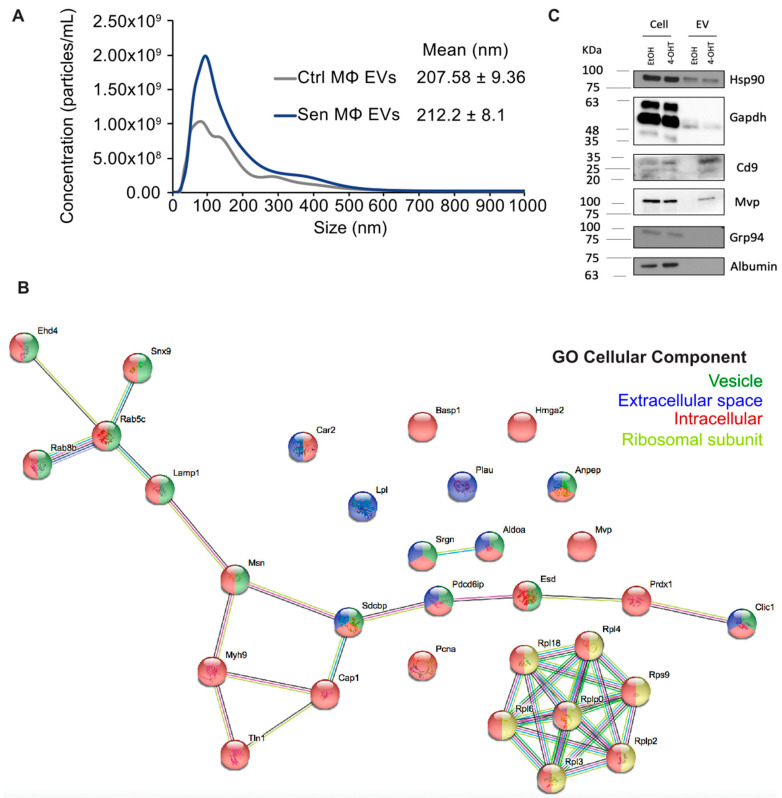
Senescent macrophages secrete more EVs carrying RNA-binding proteins and exosome proteins. (**A**) Quantification of abundance and size of EVs derived from Ctrl and Sen MΦ by fluorescent nano-particle tracking analysis (fNTA). EVs were separated from the culture medium from three independent experiments by polymer-based precipitation using Exo-Quick-TC. Mean of EV size distribution is indicated to the right. (**B**) STRING analysis of the most significant proteins regulated in EVs from Sen MΦ (Appendix A). Proteins are considered regulated when fold change ≥ |1.5| and *p*-value ≤ 0.05. Associated GO cellular components are indicated in color. (**C**) Western blots showing levels of Hsp90α/β, Cd9, Mvp, Grp94, Albumin, and Gapdh as a loading control. Protein extracts were obtained from RAW 264.7 cells expressing ∆Raf-1:ER treated for an initial 24 with 4-hydroxytamoxifen (4-OHT, 100 nM) or vehicle (EtOH), followed by a further 48 h with 4-OHT or vehicle in fresh serum-free media. The protein extracts for EV were obtained after the EVs separation from culture medium by polymer-based precipitation using Exo-Quick-TC. Experiment was performed 3 times.

**Figure 4 biomedicines-12-01089-f004:**
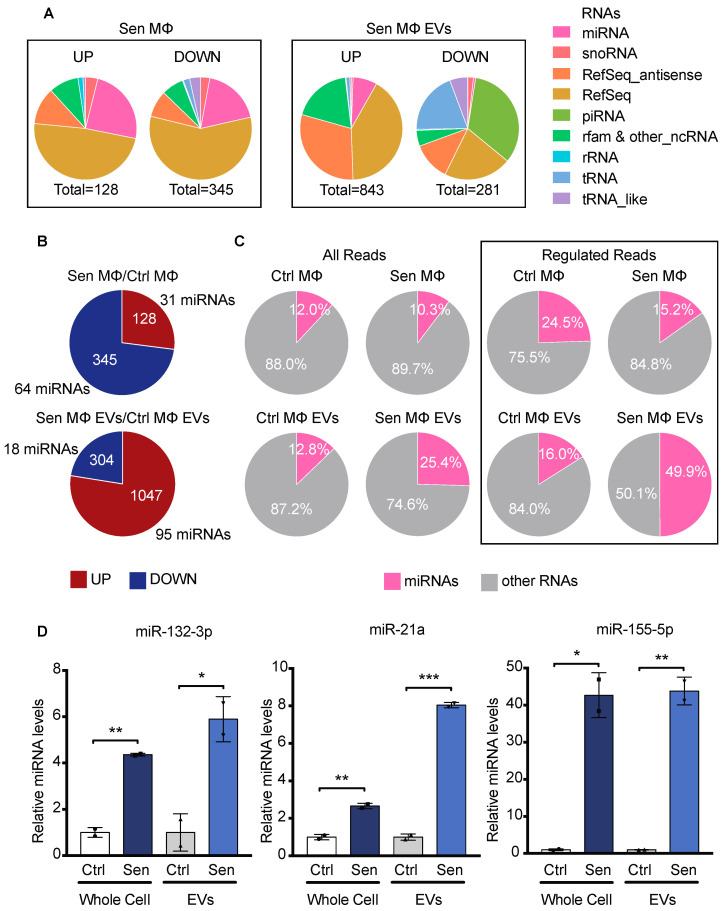
EVs from senescent macrophages carry pro-inflammatory miRNAs. (**A**) Distribution of RNA species identified as upregulated (UP) or downregulated (DOWN) with fold change ≥ |2| and *p*-adjusted ≤ 0.005 in an analysis of small RNAseq of samples of Ctrl and Sen MΦ and their secreted EVs. RefSeq refers to mRNA, and RefSeq antisense refers to the antisense transcripts of a coding sequence. (**B**) RNAs upregulated (UP) and downregulated (DOWN), identified by small RNAseq in Sen MΦ vs. Ctrl MΦ and in EVs from Sen MΦ vs. Ctrl MΦ. Fold change ≥ |2| and *p*-adjusted ≤ 0.005. The number of RNA species (i.e., tRNAs, mRNAs, rRNA, and all others) is indicated within the circles, while the number of miRNAs only is indicated outside the circle. (**C**) Proportion of reads belonging to miRNAs or to all RNA species are calculated across all small RNAseq data or within the subgroup of reads from RNAs found regulated between Sen MΦ vs. Ctrl MΦ, or EVs from Sen MΦ compared to EVs from Ctrl MΦ, using a fold change ≥ |2| and *p*-adjusted ≤ 0.005. (**D**) QPCR validation for some miRNAs enriched in EVs from Sen MΦ, as identified by small RNAseq. RNA levels were normalized over snRNA U6 and rRNA 5S. The experiment was performed 2 times. Error bars represent standard deviation; * *p*-value < 0.05; ** *p*-value < 0.01; *** *p*-value < 0.001 using Student’s *t*-test.

**Figure 5 biomedicines-12-01089-f005:**
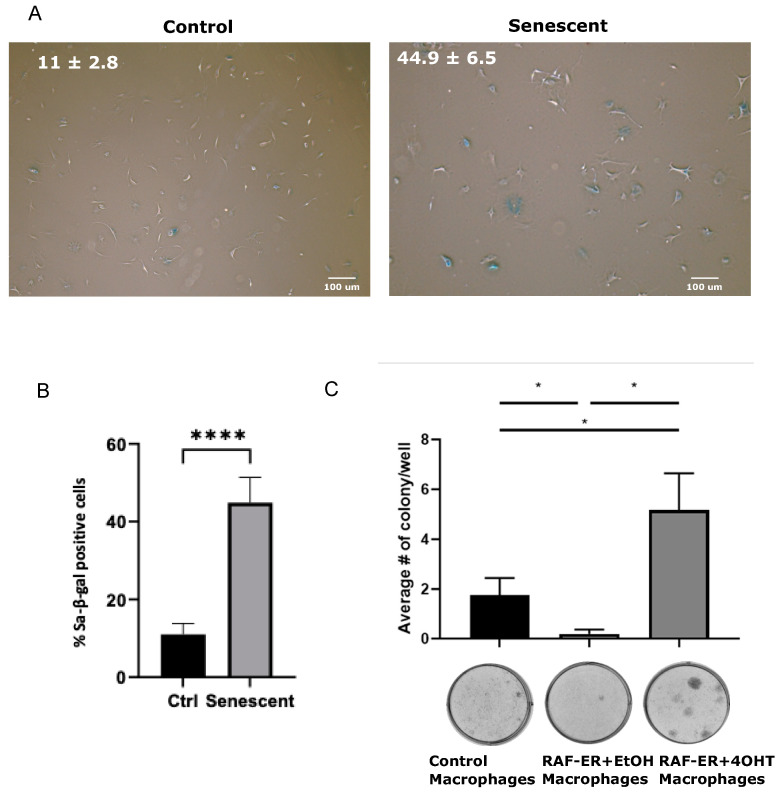
Secretions of senescent macrophages induce escape from senescence in MEFs. (**A**,**B**) Representative images (**A**) and quantification (**B**) of senescence-associated ß-Galactosidase (SAβGal) staining of young control MEFs and senescent MEFs after serial passage (10 days in culture); n = 3; error bars represent standard deviation; **** *p*-value < 0.0001 using Student’s *t*-test. (**C**) Quantification of the colony assay of senescent MEFs after 21 days of daily transfer of conditioned media from RAW 264.7 RAF-ER MΦ treated with ethanol vehicle (EtOH) or 4-OHT (100 nM). The number of colonies was quantified after staining with crystal violet, n = 3. Error bars represent standard deviation; * *p*-value < 0.05 using Student’s *t*-test. Representative plates of the colony assay are shown after staining with crystal violet at the bottom of each panel.

**Table 1 biomedicines-12-01089-t001:** Primers for SYBR Green real-time QPCR.

Specie	Gene	Forward Primer	Reverse Primer
Mus musculus	Ki67	agggtaactcgtggaaccaa	ttaacttcttggtgcatacaatgtc
	C2cd5	ggtaaaggttgtcttattcaggcaagg	ggcaagagattactgatagctgtgg
	Cntln	ggaggaagagctgagcagcctaa	ccacagagaccatacaaattccttgtc
	Patzl	cagtgggcaaaccgtacatctg	tgcacctgcttgatatgtccatt
	Trdmt1	ggttgcgagaggatggaacc	tgtgcagggatatgactttctcg
	Cdkn1a/p21	cacagctcagtggactggaa	accctagacccacaatgcag
	Pml	ccagcgtcctgccacagt	ggtgcgatatgcattcagtaactc
	Fam214b	cccaaggagcctgttttgga	tcgaagggagcttagcttcagg
	Tgfb1	gcaacatgtggaactctaccaga	acagccactcaggcgtatcag
	Pai-1	ttgtccagcgggacctagag	aagtccacctgtttcaccatagtct
	Angptl2	ccctggaggttggactgtcatc	cgatgttcccaaacccttgctt
	Tm4sf1	tgaagaggactgctgtggttgc	gggctcatagcacttggaccac
	Mcpt1	ggcacttctcttgccttctgga	catgtaaggacgggagtgtggtc
	Gm5483	gatctgccacaccagaaatcca	ggaggaaacaaccaccaccaac
	Tmem132b	tggggcccagcaaatcacct	tgcattccacagactccaacaca
	Itgb3	cgccatcatgcaggctacagt	cactagcaaatgggatgcgtca
	Slc9a2	tgggctttcgtctgctttaccc	ggtccggaaccagttaatcacc
	Upp1	acactctggaagccttctcgcg	gcacgtcttccttcattgctgct
	Itga3	gggcttgggcaaagtctacatc	cctggcagtccgagtttctctc
	Tbp	gtttctgcggtcgcgtcatttt	tctgggttatcttcacacaccatga
	β-Actin	tcctagcaccatgaagatcaagatc	ctgcttgctgatccacatctg
	SerpinE1	aggtaaacgagagcggcacagt	atgcgggctgagatgacaaag
	Plaur	tgcttcgggaatggcaagat	cctgttggtcttttcgctgtgg
	Plau	ggagcagctcatcttgcac	cccgtgctggtacgtatctt
Homo sapiens	TBP	gctggcccatagtgatctttgc	cttcacacgccaagaaacagtga
	HMBS	aacggcaatgcggctgcaa	gggtacccacgcgaatcac
	IL1A	cggttgagtttaagccaatccatc	ggtgctgacctaggcttgatga
	IL8	ggcacaaactttcagagacagca	ggcaaaactgcaccttcacaca
	IL6	ccaggagcccagctatgaactc	aaggcagcaggcaacaccag

**Table 2 biomedicines-12-01089-t002:** Primers for reverse transcription and TaqMan real-time QPCR of miRNAs.

	Primer
mmu-miR-21a-5p forward primer	gtgccgtagcttatcagactgatgttga
mmu-miR-132-3p forward primer	gtgccgtaacagtctacagccatggtcg
mmu-miR-155-5p forward primer	gtgccgttaatgctaattgtgatag
mmu-5S rRNA forward primer	ctgggaataccgggtgctgtag
mmu-U6 forward primer	cacgcaaattcgtgaagcgttccat
Universal real-time QPCR reverse primer	ccagtctcagggtccgaggtattc
Universal reverse transcription primer	cgactcgatccagtctcagggtccgaggtattcgatcc taaccctctcctcggtatcgagtcgcacttttttttttttv

## Data Availability

Data are available upon reasonable request to corresponding authors.

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
