# Peer review of "Senescent Macrophages Release Inflammatory Cytokines and RNA-Loaded Extracellular Vesicles to Circumvent Fibroblast Senescence"

_biomedicines, 2024, doi:10.3390/biomedicines12051089_

Round 1
Reviewer 1 Report
Comments and Suggestions for Authors
Camille Laliberté et al. established an in vitro model of senescence induced by Raf-1 in RAW 264.7 macrophages and analyzed the proinflammatory landscape of senescent macrophages.The work is interesting and important. The manuscript was well written.
Comments:
1.The statistical analysis method is lacking in the Methods section.
2. Fig 5D is not clear.
Minor points:
line 318: Extracellular vesicles, (EVs),--->Extracellular vesicles (EVs),
line466: RTqPCR--->RT-qPCR
line 610: QPCR can be unified with qPCR or QPCR.
supplementary material: Tubuline--->Tubulin
SA β-Gal, SA-β-Gal, SAbGal should be unified.
Comments on the Quality of English LanguageCamille Laliberté et al. established an in vitro model of senescence induced by Raf-1 in RAW 264.7 macrophages and analyzed the proinflammatory landscape of senescent macrophages.The work is interesting and important. The manuscript is well written.
Comments:
1.The statistical analysis method is lacking in the Methods section.
2. Fig 5D is not clear.
Minor points:
line 318: Extracellular vesicles, (EVs),--->Extracellular vesicles (EVs),
line466:RTqPCR--->RT-qPCRβ
line 610: QPCR can be unified with qPCR or QPCR.
supplementary material:Tubuline--->Tubulin
SA β-Gal,SA-β-Gal,SAbGal should be unified.
Author Response
"Please see the attachment."

Reviewer 2 Report
Comments and Suggestions for Authors
In the present study, an in vitro model of senescence was established, namely Raf-1 oncogene-mediated senescence in RAW 264.7 murine macrophages and a comprehensive transcriptomic analysis of senescent macrophages was conducted. Interestingly, the authors found that the secretion of senescent macrophages promoted the escape from senescence in murine embryonic fibroblasts. The study is interesting, but major revision is needed.
Specific comments:
As secretion-based escape from senescence is the most interesting finding of the study, the authors should explore the underlying mechanism(s).
The title is not informative. The title should reflect obtained results.
The abstract should be re-phrased to be more readable in terms of the novelty of the study and key findings.
Limitations of the study were also not highlighted.
Round 2
Reviewer 2 Report
Comments and Suggestions for Authors
The authors have addressed all my comments/issues.
The manuscript can be now published in Biomedicines.